# Integrated multi-omics analysis and predictive modeling of heart failure using sepsis-related gene signature

Yiping Lang[1], Tianyu Liang[2], Fei Li[1]*

1 Department of Nursing, Emergency and Critical Care Center, Intensive Care Unit, Zhejiang Provincial People's Hospital, Affiliated People's Hospital, Hangzhou Medical College, Hangzhou, Zhejiang, China, 2 Emergency and Critical Care Center, Intensive Care Unit, Zhejiang Provincial People's Hospital, Affiliated People's Hospital, Hangzhou Medical College, Hangzhou, Zhejiang, China

* lifeilifeiicu@163.com

## Abstract

### Background

Heart failure (HF) is characterized by complex molecular alterations, and recent studies suggest a potential role for sepsis-related genes in cardiovascular dysfunction. This study aimed to develop a predictive model for HF based on sepsis-related gene signatures.

### Methods

Three sepsis-related datasets (GSE65682, GSE54514, and GSE95233) were analyzed to identify differentially expressed genes (DEGs) following batch effect correction using the ComBat algorithm. With the use of elastic net regularization and the glmnet package in R, Lasso Cox regression was employed to screen out gene signatures. A predictive model was developed based on the expression of each gene signature and the co-efficient values. In addition, the predictive model was validated on independent HF datasets (GSE57345, GSE141910, and GSE5406). Model performance was assessed through receiver operating characteristic (ROC) analysis and AUC values of each gene signature, and immune infiltration was evaluated using CIBERSORT, IPS, and xCell. Sepsis models of C57BL/6 mice were established by cecal ligation and puncture (CLP).

### Results

We identified 340 up-regulated and 333 down-regulated sepsis-related genes. The predictive model, incorporating six key genes, demonstrated superior performance compared to individual genes across both training and validation datasets with the AUC value of the risk score above 0.9, significantly higher than that of a single gene. Immune infiltration profiles differed significantly between HF patients and controls,

**Data availability statement:** Data is provided within the manuscript or supporting information files. All the raw data can be found at the following link: https://doi.org/10.57760/sciencedb.23650.

**Funding:** The author(s) received no specific funding for this work.

**Competing interests:** The authors have declared that no competing interests exist.

**Abbreviations:** HF, Heart failure; DEG, Differentially expressed gene; EF, Ejection fraction; BNP, Brain natriuretic peptide; NT-proBNP, N-terminal pro-b-type natriuretic peptide; MRAs, Mineralocorticoid receptor antagonists; ET-1, Endothelin-1; PPI, Protein-protein interaction; PCA, Principal Component Analysis; GEO, Gene Expression Omnibus; GO, Gene Ontology; KEGG, Kyoto Encyclopedia of Genes and Genomes; FDR, False discovery rate; ROC, Receiver operating characteristic; HFrEF, HF with reduced ejection fraction; ROS, Reactive oxygen species; NO, Nitric oxide; CBV3, Coxsackievirus B3; HLA, Human leukocyte antigen; CLP, Cecal ligation and puncture; cDNA, Complementary DNA; qRT-PCR, Quantitative reverse transcription polymerase chain reaction.

with more pronounced alterations observed at higher risk score levels. Finally, the expression of six key genes in sepsis models was confirmed to be consistent with our prediction.

## Conclusion

The model constructed through sepsis-related characteristic genes provides a highly advantageous method for predicting HF, and the characteristic genes we have screened may be potential biomarkers for predicting HF. This model has potential application value in early diagnosis and risk stratification, which can help improve the clinical management of heart failure and provide new ideas for preventing HF.

## 1. Introduction

Sepsis is defined as a systemic inflammatory response to infection, often leading to multiple organ dysfunction, and in severe cases, progressing to septic shock. Recent updates to the diagnostic criteria for sepsis have influenced its epidemiological characteristics and clinical identification [1,2]. This systemic inflammatory burden not only affects multiple organs but also has profound implications for cardiac function. During sepsis, the inflammatory response causes an excessive production of catecholamines, which impairs myocardial function and contractility. Furthermore, when tachycardia leads to decreased coronary perfusion and cardiac output, overall cardiac output is negatively impacted [3]. Both sepsis and heart failure (HF) significantly contribute to the incidence of various complications and mortality rates. The mortality rate associated with septic shock is approximately 40% [4], while the five-year mortality rate for diagnosed HF is around 50% [5]. Notably, sepsis accounts for one-quarter of all deaths among HF patients, thereby establishing a crucial link between the occurrence or exacerbation of HF and sepsis. HF is a syndrome characterized by the heart's inability to pump sufficient blood and oxygen to meet the metabolic demands of other organs. HF is a prevalent cardiovascular disorder that impacts the health of millions worldwide [6]. Ejection fraction (EF) is commonly used as a prognostic indicator and for the classification of HF patients, serving as a crucial clinical parameter in this context [7]. HF currently impacts approximately 64 million people worldwide, with increasing prevalence attributed to an aging population, a rising burden of HF comorbidities and risk factors, and improved survival following myocardial infarction. In clinical practice, B-type natriuretic peptide (BNP) and N-terminal pro-b-type natriuretic peptide (NT-proBNP) are regarded as gold standards for the prognostic diagnosis and stratification of HF [8]. The early use of beta-blockers and mineralocorticoid receptor antagonists (MRAs) has been shown to reduce the incidence of composite endpoints during hospitalization, as well as overall mortality, in-hospital mortality, and rates of readmission following discharge [9,10].

Sepsis significantly impairs cardiac function and often contributes to the onset of heart failure. Research has shown that impaired cardiac function in septic patients is associated with elevated levels of various biomarkers, including BNP and

endothelin-1 (ET-1) [11,12]. Moreover, another study revealed that septic patients with concurrent heart failure exhibit significantly higher serum levels of CXCL8 and ET-1 compared to those with sepsis alone, suggesting that these biomarkers may play important roles in the pathophysiology of sepsis complicated by HF [12]. Given the substantial clinical overlap between sepsis and HF, understanding the molecular mechanisms linking these two conditions is vital for improving patient outcomes. Despite advancements in treatment strategies, reliable predictive models are still needed to identify the risk of HF in septic patients and to provide insights into potential therapeutic targets.

Bioinformatics enables the systematic analysis of large-scale gene expression data, revealing the complex molecular mechanisms underlying diseases [13,14]. Through the application of bioinformatics analytical methods, various biomarkers regulating the progression of HF have been identified [15–17]. Moreover, in the context of sepsis and heart failure, bioinformatics plays a crucial role in identifying sepsis-related biomarkers associated with cardiac dysfunction and disease progression [18]. Common bioinformatics approaches utilized include differential gene expression analysis, protein-protein interaction (PPI) network construction, and pathway enrichment analysis, all of which aim to provide valuable insights into the biological processes linking sepsis and heart failure. Additionally, by integrating multi-omics data, these methods facilitate the development of predictive models that can identify key genes and pathways related to both conditions [19]. This approach not only enhances our understanding of the molecular interactions between sepsis and heart failure but also highlights potential biomarkers for early diagnosis and novel therapeutic targets for intervention.

In this study, we performed a multi-dataset analysis and applied elastic net regularization to develop a predictive model based on sepsis-related gene expression profiles to predict the occurrence of HF. Furthermore, we analyzed the immune infiltration patterns in HF patients to explore how variations in immune profiles may be associated with disease progression. Through comprehensive bioinformatic analysis and validation across independent datasets, we aim to enhance the understanding of the molecular mechanisms linking sepsis-related pathways to HF and to provide reliable tools for early diagnosis and risk stratification.

## 2. Methods

### 2.1. Data acquisition, preprocessing, and batch effect correction

We retrieved three sepsis-related gene expression datasets from the Gene Expression Omnibus (GEO): GSE65682, GSE54514, and GSE95233, which included samples from both sepsis patients and healthy controls. To ensure data uniformity, we conducted preprocessing that involved normalization and log transformation, making the datasets amenable for comparative analysis. Batch effects, inherent in the integration of multiple datasets, were initially assessed using Principal Component Analysis (PCA) and density plots, revealing dataset-specific clustering and distribution variations. To address these batch effects, we employed the 'Combat' algorithm from the 'sva' package in R, which standardized means and variances across datasets, thereby enhancing data comparability for subsequent analyses [19]. We then performed differential expression analysis to identify key genes dysregulated in sepsis versus controls, with a significance threshold set at $p < 0.05$. Volcano plots were created to visualize the distribution of differentially expressed genes, illustrating both up-regulated and down-regulated genes in each dataset.

### 2.2. Identification and dimensionality reduction of sepsis-related gene expression patterns

Three datasets (GSE65682, GSE54514, and GSE95233) were retrieved from the GEO database to identify sepsis-related genes. Venn diagrams were employed to determine the overlap of up-regulated and down-regulated genes across datasets. To effectively visualize gene expression differences, we calculated the mean expression and standard error of the up-regulated and down-regulated genes within the same groups. PCA was performed to evaluate the inter-cohort distances, offering a summary of the overall variance in gene expression. Additionally, we normalized the expression profiles for each dataset using z-score transformation and applied the 'prcomp' function to perform dimensionality reduction.

Finally, a heatmap was generated to illustrate the expression levels of sepsis-related genes across all samples, providing insights into inter-individual variability.

## 2.3. PPI network construction and module analysis of sepsis-related genes

We utilized the STRING database (https://cn.string-db.org/) to construct PPI networks for the differentially expressed sepsis-related genes. STRING is a widely used platform for integrating and visualizing known and predicted protein-protein interactions, facilitating the exploration of cellular pathways and molecular mechanisms. Separate PPI networks were generated for up-regulated and down-regulated sepsis-related genes to uncover interaction patterns relevant to the pathogenesis of sepsis. To further investigate potential functional modules within these networks, we employed the 'dynamicTreeCut' algorithm in R, setting the minimum module size to two genes. This approach enabled the identification of cohesive gene clusters that may share related biological functions or participate in common signaling pathways.

## 2.4. GO and KEGG pathway enrichment analysis of sepsis-related genes

Gene Ontology (GO) enrichment analysis was conducted to investigate the biological processes, cellular components, and molecular functions associated with up-regulated and down-regulated sepsis-related genes. Separate GO analyses were performed for both gene groups to identify functional distinctions. To further explore the involvement of these genes in signaling pathways, we carried out the Kyoto Encyclopedia of Genes and Genomes (KEGG) pathway enrichment analysis. For improved clarity and interpretability, similar KEGG terms were merged, providing a concise overview of the enriched pathways. GO and KEGG enrichment analyses revealed several significantly enriched biological pathways. Pathways with a p-value < 0.05 and false discovery rate (FDR) < 0.05 were considered statistically significant.

## 2.5. Development of a predictive model for HF using sepsis-related genes

We developed a predictive model for HF using three publicly available datasets: GSE57345, GSE141910, and GSE5406, focusing on sepsis-related genes. The elastic net regularization method, which combines ridge regression and Lasso, was utilized to fit a generalized linear model using the 'glmnet' package in R [20]. To ensure data consistency, batch effects across datasets were corrected with the 'ComBat' algorithm, retaining only common genes for further analysis. The elastic net penalty parameter was set to $\alpha = 0.9$, balancing Lasso and ridge regression. Cross-validation within the training dataset was employed to identify the optimal regularization parameter, lambda, which dictates the degree of shrinkage in model fitting [21]. Furthermore, a leave-one-study-out validation method was used to assess the model's robustness across different training subsets. Genes selected by the elastic net were then integrated using logistic regression to create the final predictive model, which was evaluated on testing datasets to confirm its reliability in predicting HF.

## 2.6. Differential expression analysis, ROC analysis, and PPI network construction of model genes

We performed differential expression analysis on the six model genes to compare their expression between HF and control samples across both training and validation datasets. Receiver operating characteristic (ROC) curve analysis was used to evaluate the predictive performance of the constructed risk score relative to individual model genes. Immune checkpoint genes were found to be negatively correlated with the risk score. To provide a more detailed breakdown, we compared the expression levels of immune checkpoint genes between the high- and low-risk groups. Several checkpoint molecules, including PDCD1 (PD-1), CD274 (PD-L1), CTLA4, LAG3, and others, were significantly changed in the high-risk group. These results are illustrated in S1 Fig. Additionally, we explored potential PPIs involving the model genes by identifying interacting proteins and calculating cumulative interaction scores based on multiple layers of supporting evidence. The strength of each interaction was represented by the length of the connecting lines, with shorter lines indicating stronger evidence. Proteins positioned closer to the center of the network were assigned higher interaction confidence, reflecting greater biological relevance.

## 2.7. Immune infiltration quantification and correlation analysis

We utilized three immune infiltration quantification methods—CIBERSORT, IPS, and xCell—to comprehensively assess immune cell composition and immune-related activity within the training dataset. These complementary approaches provided detailed insights into immune landscape variations associated with HF. Additionally, correlation analyses were performed to evaluate the associations between the risk score and 46 immune-related metrics. To further explore immune regulatory dynamics, we assessed the correlation between the risk score and the expression of immune checkpoint genes, as well as its relationship with human leukocyte antigen (HLA) family gene expression, offering insights into antigen presentation and immune modulation.

## 2.8. Sepsis mouse model

All animal experiments conducted in this study were approved by the Institutional Animal Care and Use Committee of the Hubei Provincial Center for Disease Control and Prevention. Male C57BL/6 mice (8−10 weeks) were subjected to cecal ligation and puncture (CLP) to induce sepsis. All mice were anaesthetised with a solution of 3% pentobarbital sodium. Subsequently, a 2-cm midline incision was made to expose the cecum using a scalpel. Then one-third of the ileum-cecum junction was ligated with 4−0 silk thread and punctured twice with an 18 G needle. Twenty-four hours after the successful construction of the model, the mice were euthanised, and the tissue samples were collected. During the euthanasia of the mice, we initially gave a sedative (5 mg/kg) through an intraperitoneal injection. Once the mice became calm and unconscious, we administered a lethal dose of pentobarbital sodium (150 mg/kg) via intravenous injection. We then observed the heart rate, respiration, and pupillary reflexes of the mice to verify their death shortly thereafter.

## 2.9. RNA extraction and qRT-qPCR

Total RNA was extracted from tissue samples using TRIzol (Invitrogen, USA) and converted to complementary DNA (cDNA) using a cDNA synthesis kit (Vazyme, China). The quantitative reverse transcription polymerase chain reaction (qRT-qPCR) was performed using SYBR Green qRT-PCR Master Mix (Vazyme China) and with GAPDH as the internal control. Primers used are listed in Supplementary Table S1.

## 2.10. Ethics approval and consent to participate

The animal experiment was approved by the Institutional Animal Care and Use Committee of the Hubei Provincial Center for Disease Control and Prevention. There are no human subjects in this article and informed consent is not applicable.

## 2.11. Statistical analysis

All statistical analyses of bioinformatics data were performed using R software (version 4.1.2). A significance level of $p < 0.05$ was deemed statistically significant. For cellular experimental data analysis, GraphPad Prism 9.0 software (GraphPad, San Diego, CA) was utilized. All experiments were conducted in triplicate, and results were expressed as mean ± standard deviation (SD). Statistical comparisons between the two groups were conducted using a t-test, with p-values below 0.05 considered statistically significant. To ensure result reproducibility, all experiments were repeated a minimum of three times.

# 3. Results

### 3.1. Multi-dataset integration and identification of sepsis-related genes

The sample distribution across the three datasets is presented in the petal diagram (Fig 1A), which delineates the proportions of sepsis and control samples, facilitating an overview of dataset composition. An UpSet plot (Fig 1B) illustrates the intersection and uniqueness of gene sets across the datasets, revealing overlaps that provide insights into shared

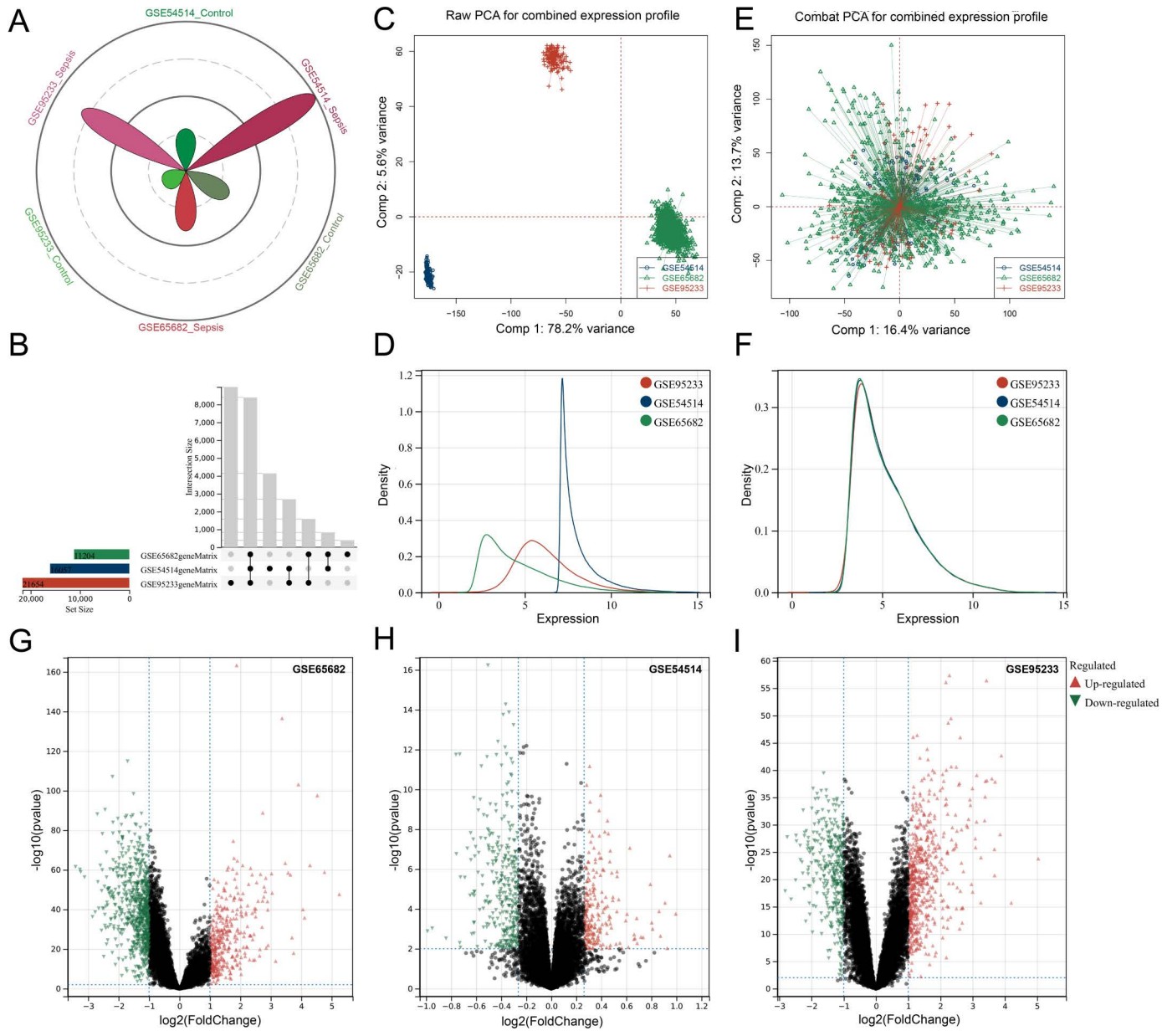

**Fig 1. Batch Effect Correction of Multi-Dataset Integration and Differential Expression Analysis.** (A) Petal diagram illustrating the sample size distribution for sepsis and control groups across GSE65682, GSE54514, and GSE95233. (B) UpSet plot depicting the intersection and unique gene sets among the three datasets. (C) PCA plot before batch correction, showing dataset-specific clustering, indicating batch effects. (D) Density plot illustrating significant differences in data distribution across datasets before batch correction. (E) PCA plot after Combat correction, demonstrating mixed clustering of samples from different datasets, confirming successful batch effect removal. (F) Density plot after batch correction, showing aligned data distributions with comparable means and variances across datasets. (G-I) Volcano plots for GSE65682 (G), GSE54514 (H), and GSE95233 (I), visualizing the up-regulated (red) and down-regulated (green) genes with p-value < 0.05.

molecular signatures. PCA analysis before batch correction (Fig 1C) shows distinct clustering of samples by dataset, indicating the presence of batch effects, while the density plot (Fig 1D) further corroborates these findings by demonstrating significant differences in data distributions. Following Combat correction through the scale function of the R package to

normalize, the PCA plot (Fig 1E) confirms successful integration, as samples from different datasets now cluster together, suggesting the removal of batch effects. Additionally, the density plot after correction (Fig 1F) displays aligned distributions with comparable means and variances, further supporting the effectiveness of the batch effect correction. Differential expression analysis revealed a range of sepsis-associated genes, with volcano plots for GSE65682 (Fig 1G), GSE54514 (Fig 1H), and GSE95233 (Fig 1I) displaying the up-regulated genes in red and the down-regulated genes in blue. These results emphasize both the shared and unique molecular patterns across datasets, enhancing our understanding of the gene expression alterations that characterize sepsis.

### 3.2. Expression patterns and cohort clustering of sepsis-related genes

The Venn diagrams reveal the shared gene sets across datasets, identifying 340 up-regulated genes (Fig 2A) and 333 down-regulated genes (Fig 2B). The PCA plot based on the mean expression and standard error (Fig 2C) demonstrates a clear separation between sepsis patients and healthy controls, highlighting distinct molecular profiles between these groups. Dimensionality reduction using z-score normalization followed by PCA (Figs 2D-F) confirms that sepsis

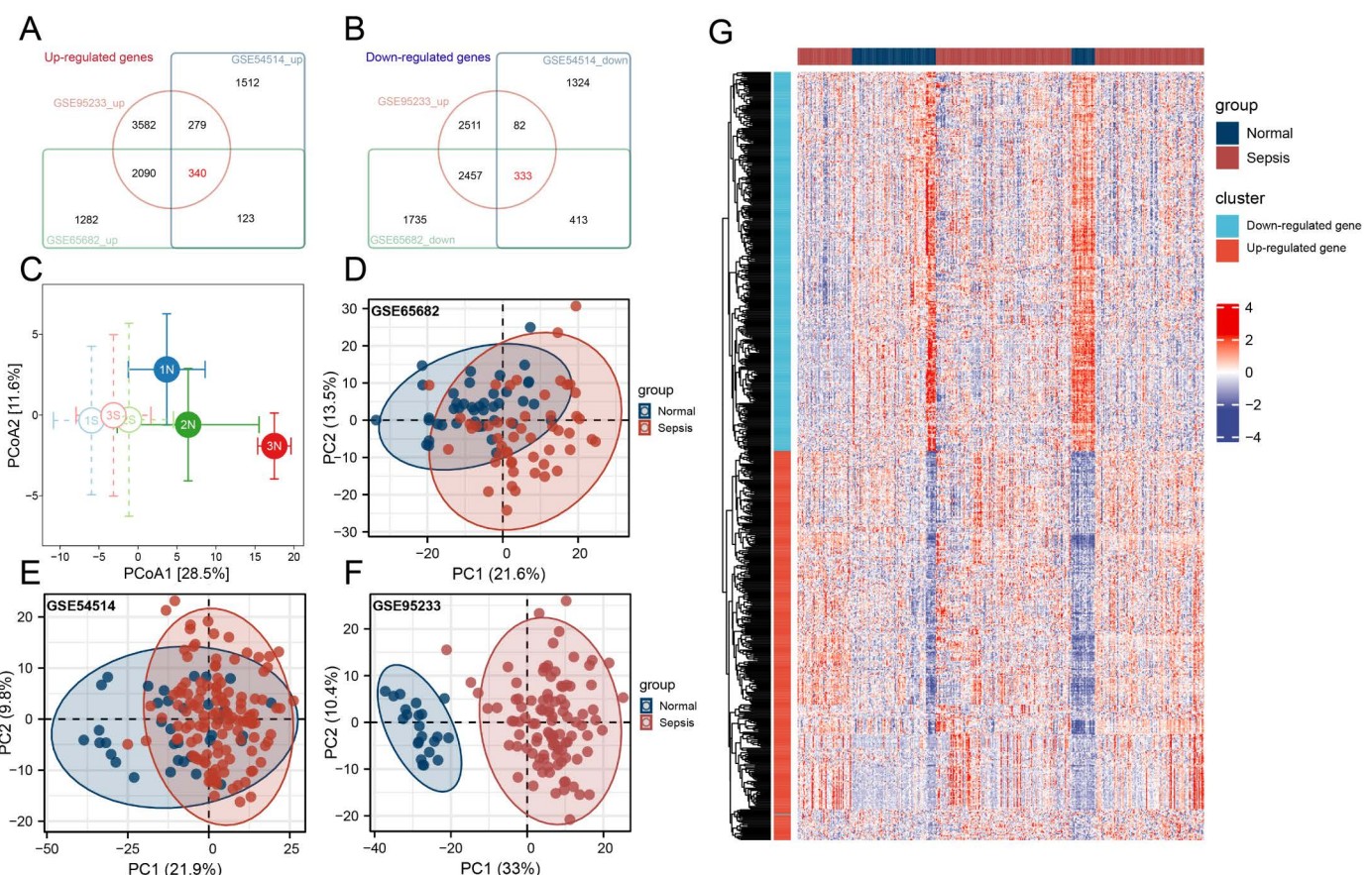

**Fig 2. Identification, Dimensionality Reduction, and Expression Visualization of Sepsis-Related Genes.** (A) Venn diagram showing the 340 up-regulated sepsis-associated genes common across the three datasets. (B) Venn diagram displaying the 333 down-regulated sepsis-associated genes shared across the datasets. (C) PCA plot illustrating the inter-cohort distances based on the mean expression and standard error of sepsis-related genes, demonstrating distinct separation between sepsis and control samples. (D-F) PCA plots for individual datasets (GSE65682, GSE54514, and GSE95233) after z-score normalization, showing distinct clustering of sepsis and control samples. (G) Heatmap of sepsis-associated gene expression across all samples, highlighting marked inter-patient variability.

and control samples form distinct clusters within each dataset, preserving the biological differences. The heatmap of sepsis-related gene expression (Fig 2G) shows substantial inter-patient variability, reflecting the heterogeneous nature of sepsis at the molecular level.

### 3.3. PPI network insights and module identification of sepsis-related genes

The PPI network of up-regulated sepsis-related genes, shown in Fig 3A, reveals critical molecular interactions potentially contributing to sepsis pathogenesis. Similarly, the PPI network for down-regulated genes, presented in Fig 3B, highlights distinct interaction patterns among the identified proteins. Using the 'dynamicTreeCut' algorithm, we identified six functional modules within the PPI networks (Fig 3C), suggesting that genes within each module may be involved in related biological processes or participate in coordinated regulatory pathways. These findings provide further insights into the molecular architecture underlying sepsis.

### 3.4. GO and KEGG enrichment results of sepsis-related genes

GO enrichment analysis of the up-regulated sepsis-related genes demonstrated significant enrichment in pathways such as mitotic nuclear division, secretory granule, mitochondrial respirasome, and enzyme binding (Fig 4A). Conversely, the GO analysis of down-regulated genes revealed enrichment in pathways related to the regulation of metabolic processes, protein binding, nucleoplasm, and intracellular anatomical structures (Fig 4B). The KEGG pathway analysis provided complementary insights by merging similar terms, offering a comprehensive summary of the enriched pathways. In the merged KEGG analysis (Fig 4C), pathways enriched by up-regulated genes are represented by red bars, while those enriched by down-regulated genes are shown in blue, reflecting their distinct biological roles.

### 3.5. Performance of the predictive model for HF using sepsis-related genes

Using the regularization parameter that minimized binomial deviance, we developed a binomial classifier from all samples in the training datasets (Fig 5A). The classifier was constructed based on the expression profiles of six key genes identified through elastic net regression (Fig 5B). To explore the relationship between these genes across datasets, we analyzed the correlation of their expression between the training and testing datasets (Fig 5C), providing insight into their expression patterns. An analysis of the genes with non-zero coefficients indicated that their expression patterns were nearly mutually exclusive (Fig 5D), suggesting that each gene contributed independently to the predictive model. To validate the performance of the classifier, we tested it on two independent datasets, GSE141910 and GSE5406. As shown in Fig 5E, the model effectively distinguished HF patients from healthy controls, demonstrating its predictive capacity across multiple datasets.

### 3.6. Differential expression, ROC analysis, and PPI results of model genes

The differential expression analysis revealed that all six model genes exhibited significant expression differences between HF and control samples in the training dataset (Fig 6A), with these findings consistently validated in the independent validation dataset (Fig 6B). ROC curve analysis demonstrated that the risk score achieved superior predictive performance for HF compared to any individual gene in the training dataset with the highest AUC value, which indicates the high sensitivity and specificity of the risk score. (Fig 6C). Similar results were observed in both independent validation datasets, confirming the robustness of the risk score (Figs 6DE). The PPI network analysis identified several interacting proteins for the model genes, with three genes—GNMT, FURIN, and BEX1—exhibiting experimentally supported interactions (Figs 6F-H). Interaction scores were calculated to reflect the strength of supporting evidence, with shorter lines representing stronger interactions. Proteins located closer to the center of the network were assigned higher interaction confidence, indicating their biological significance.

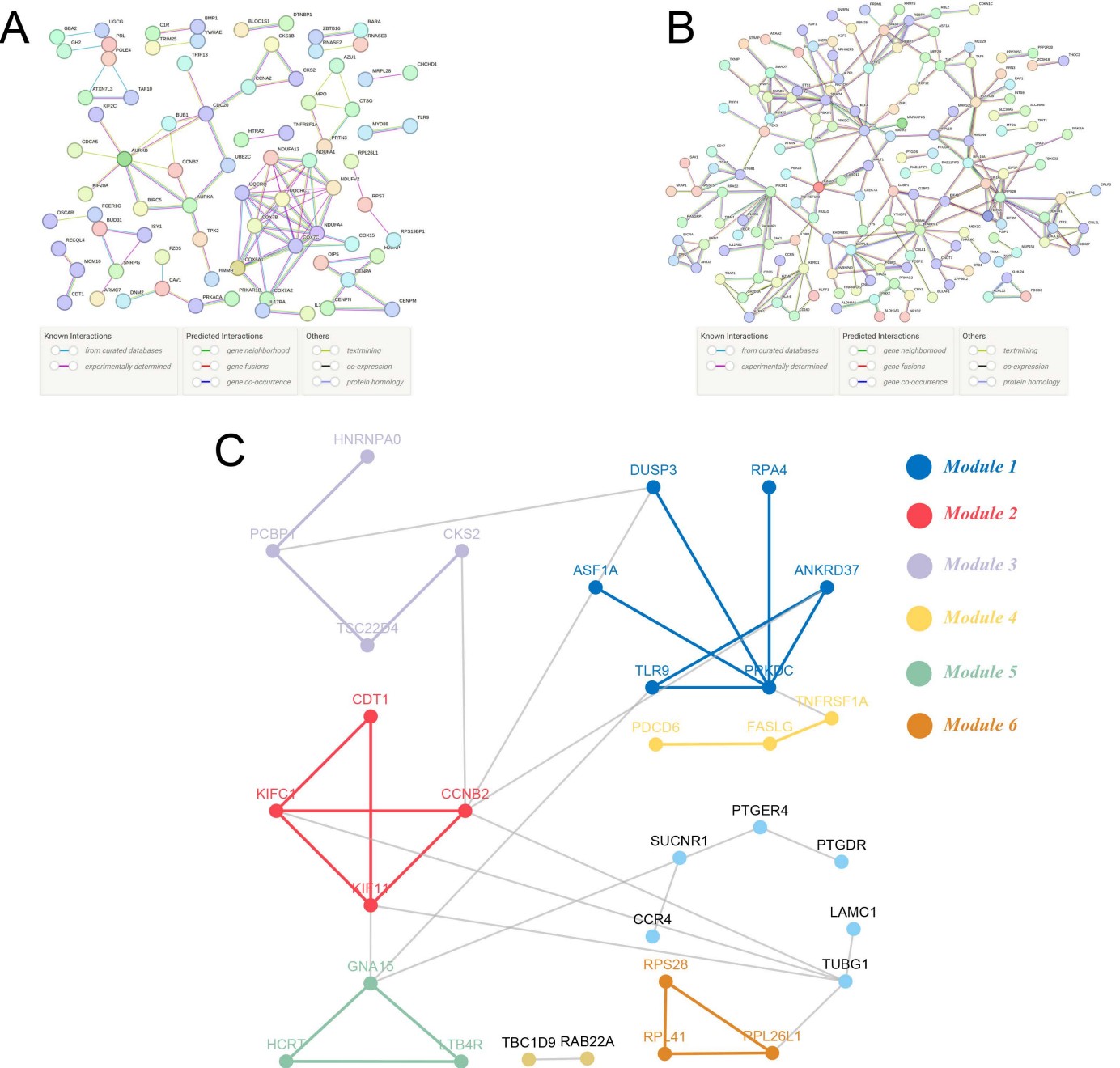

**Fig 3. PPI Network and Module Identification of Sepsis-Related Genes.** (A) PPI network of up-regulated sepsis-related genes, constructed using the STRING database, illustrating key molecular interactions and their potential roles in sepsis pathogenesis. (B) PPI network of down-regulated sepsis-related genes, visualizing distinct functional interactions among the proteins involved. (C) Module analysis of the PPI networks using the 'dynamicTree-Cut' algorithm, identifying six distinct gene modules. Each module contains at least two genes, indicating functional coherence and potential relevance to shared biological processes.

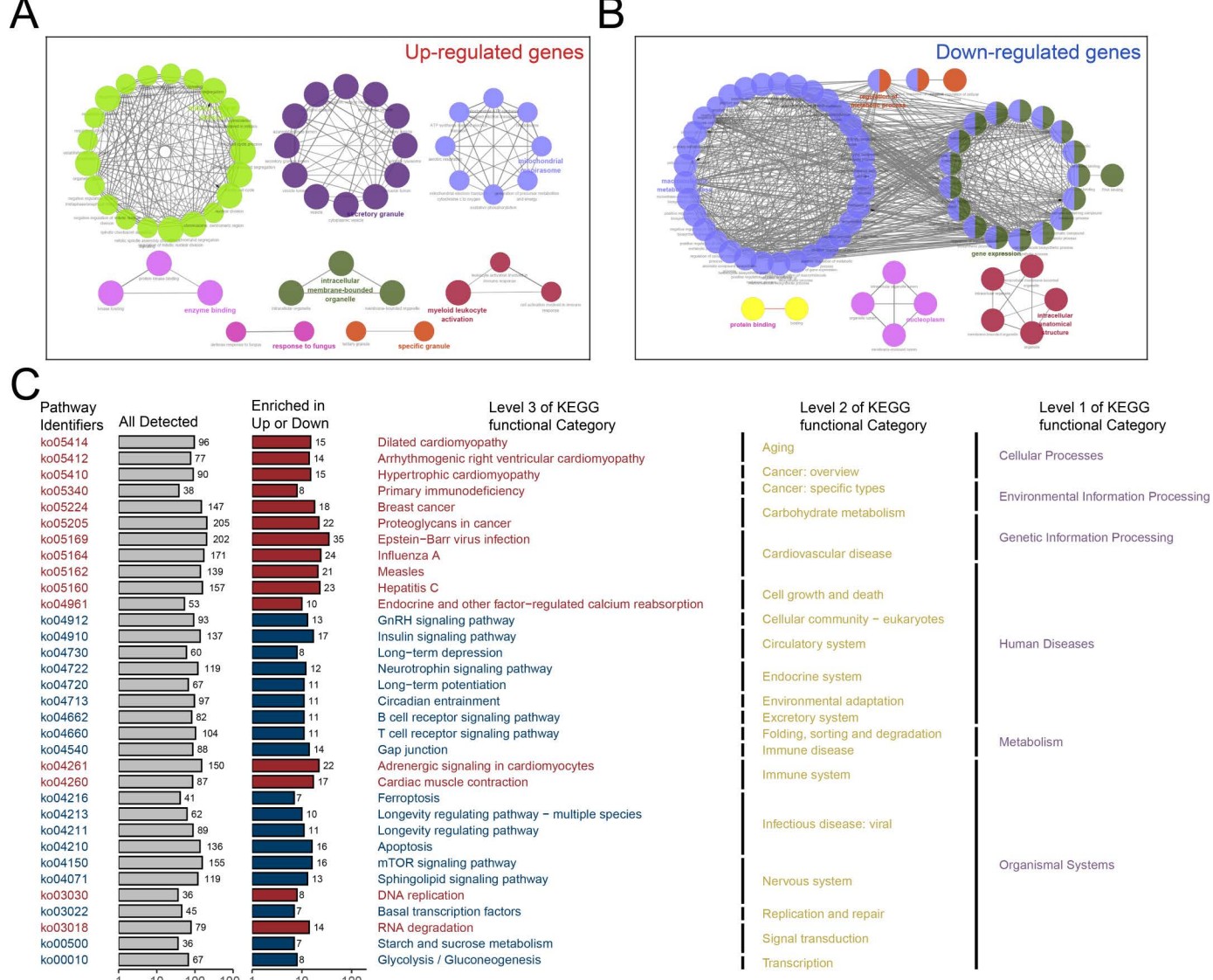

**Fig 4. GO and KEGG Pathway Enrichment Analysis of Sepsis-Related Genes.** (A) GO enrichment analysis of up-regulated sepsis-related genes, highlighting enrichment in pathways such as mitotic nuclear division, secretory granule, mitochondrial respirasome, and enzyme binding. (B) GO enrichment analysis of down-regulated sepsis-related genes, showing enrichment in pathways including regulation of metabolic processes, protein binding, nucleoplasm, and intracellular anatomical structures. (C) Merged KEGG pathway enrichment analysis, with red bars representing pathways enriched by up-regulated genes and blue bars indicating pathways enriched by down-regulated genes.

## 3.7. Immune infiltration results and association with risk score

Immune infiltration analysis revealed significant differences in immune metrics between HF patients and healthy controls, with these differences becoming more pronounced at higher levels of the risk score (Fig 7A). Correlation analysis demonstrated that the risk score was significantly associated with most of the 46 immune-related metrics, indicating its broad impact on the immune landscape (Fig 7B). Furthermore, a significant negative correlation was observed between the risk score and immune checkpoint gene expression, suggesting impaired immune regulation (Fig 7C). In contrast, the risk score exhibited a significant positive correlation with the expression of HLA family genes, indicating enhanced antigen presentation capacity in individuals with higher risk scores (Fig 7D).

                                    

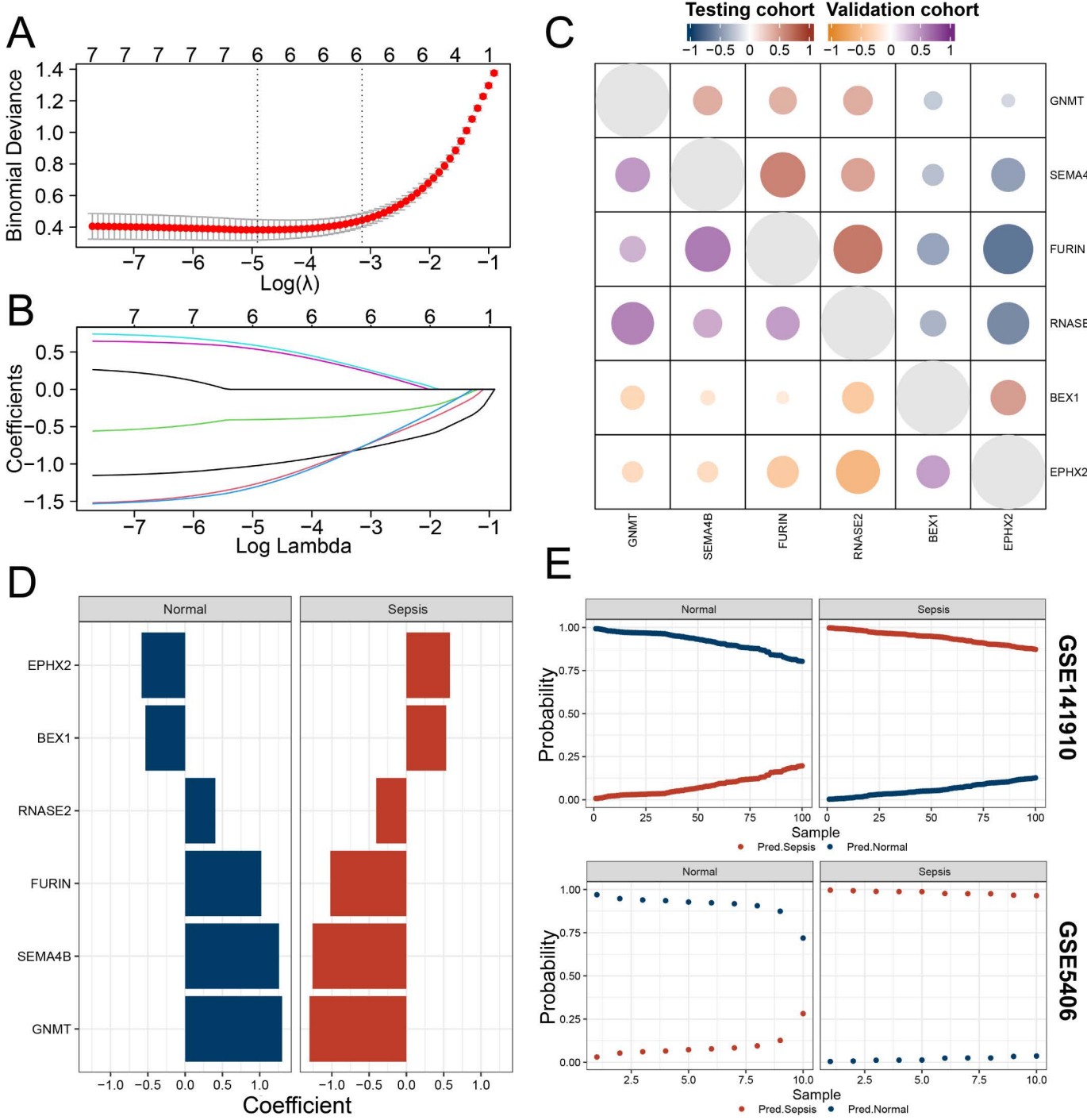

**Fig 5. Performance and Validation of the Predictive Model for HF.** (A) Development of the binomial classifier using the regularization parameter that minimized binomial deviance in the training datasets. (B) Expression profiles of the six key genes used to construct the predictive model. (C) Correlation analysis of the expression levels of the six model genes between the training and testing datasets. (D) Analysis of genes with non-zero coefficients, showing that their expression profiles are nearly mutually exclusive. (E) Validation of the predictive model on independent datasets GSE141910 and GSE5406, confirming the model's ability to differentiate HF patients from healthy controls.

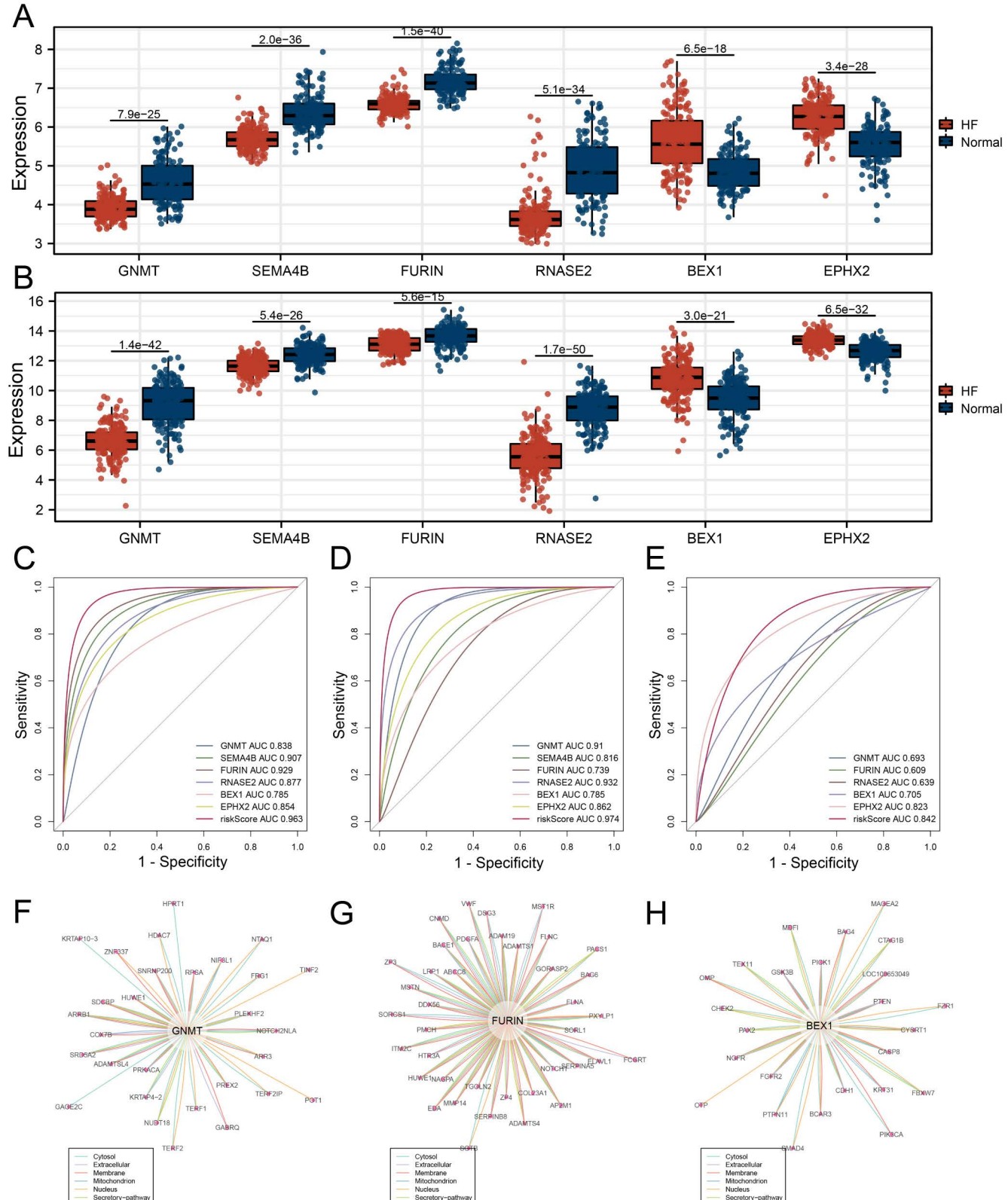

**Fig 6. Differential Expression, ROC Analysis, and PPI Network of Model Genes.** (A) Differential expression analysis of the model genes in the training dataset, showing significant differences between HF and control samples. (B) Differential expression analysis in the validation dataset, confirming

the significant expression patterns observed in the training dataset. (C) ROC curve analysis in the training dataset, demonstrating that the risk score provides superior predictive performance for HF compared to individual model genes. (D-E) ROC curve analyses in two independent validation datasets, showing that the risk score consistently outperforms individual genes in predicting HF. (F-H) PPI network analysis of GNMT, FURIN, and BEX1, with interactions supported by experimental evidence. The length of the lines reflects the strength of interaction evidence, with shorter lines indicating stronger interactions.

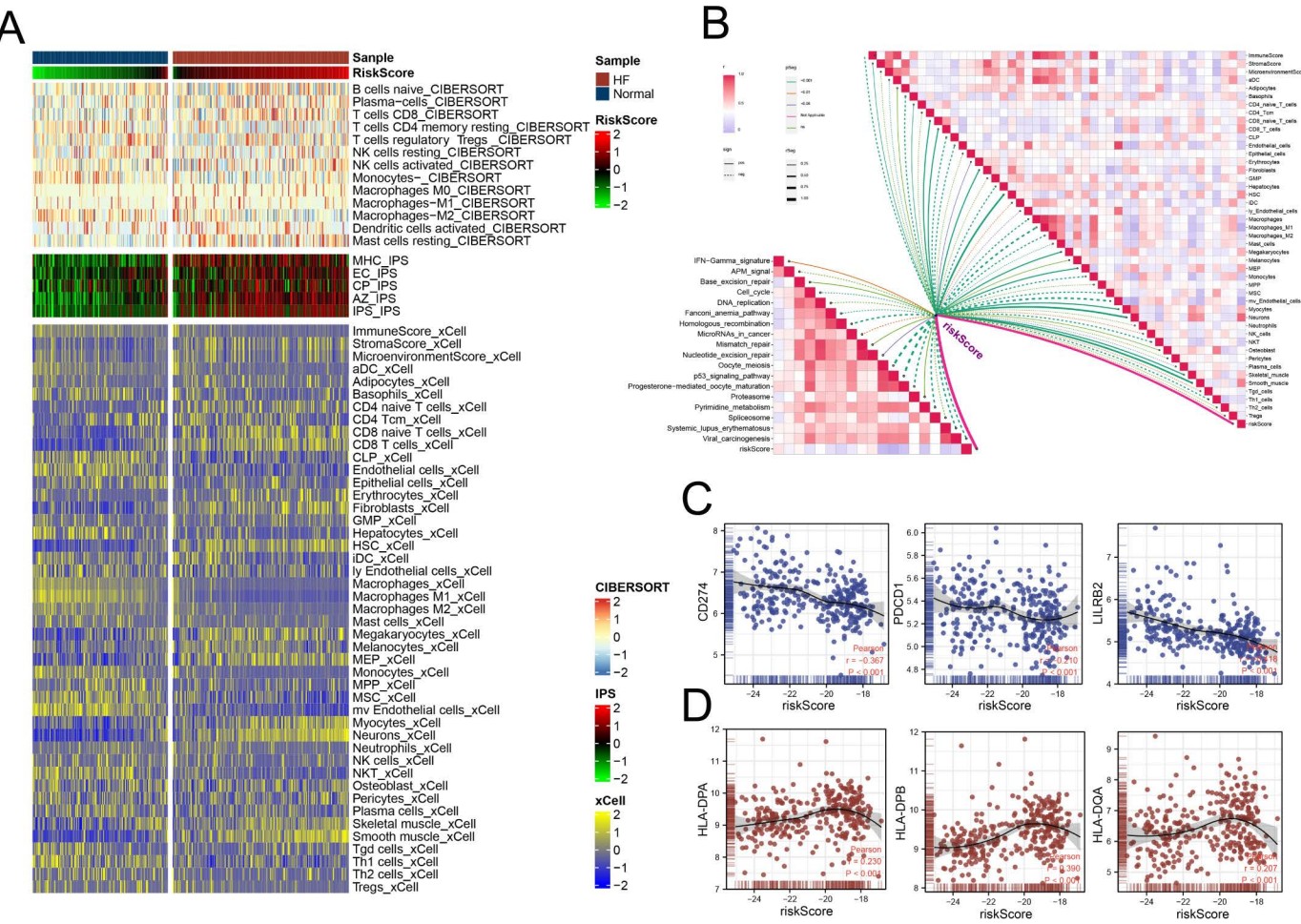

**Fig 7. Immune Infiltration and Association with Risk Score in HF Patients.** (A) Immune infiltration analysis using CIBERSORT, IPS, and xCell, showing significant differences between HF patients and control samples, with more pronounced differences at higher risk score levels. (B) Correlation analysis between the risk score and 46 immune-related metrics, demonstrating significant associations. (C) Negative correlation between the risk score and immune checkpoint gene expression. (D) Positive correlation between the risk score and HLA family gene expression.

### 3.8. The expression levels of six key genes in sepsis models

As shown in Fig 8, the relative expression levels of GNMT, SEMA4B, FURIN, and RNASE2 were significantly lower in the sepsis group than those in the control group. However, the relative expression levels of BEX1, and EPHX2 were significantly higher in the sepsis group than those in the control group.

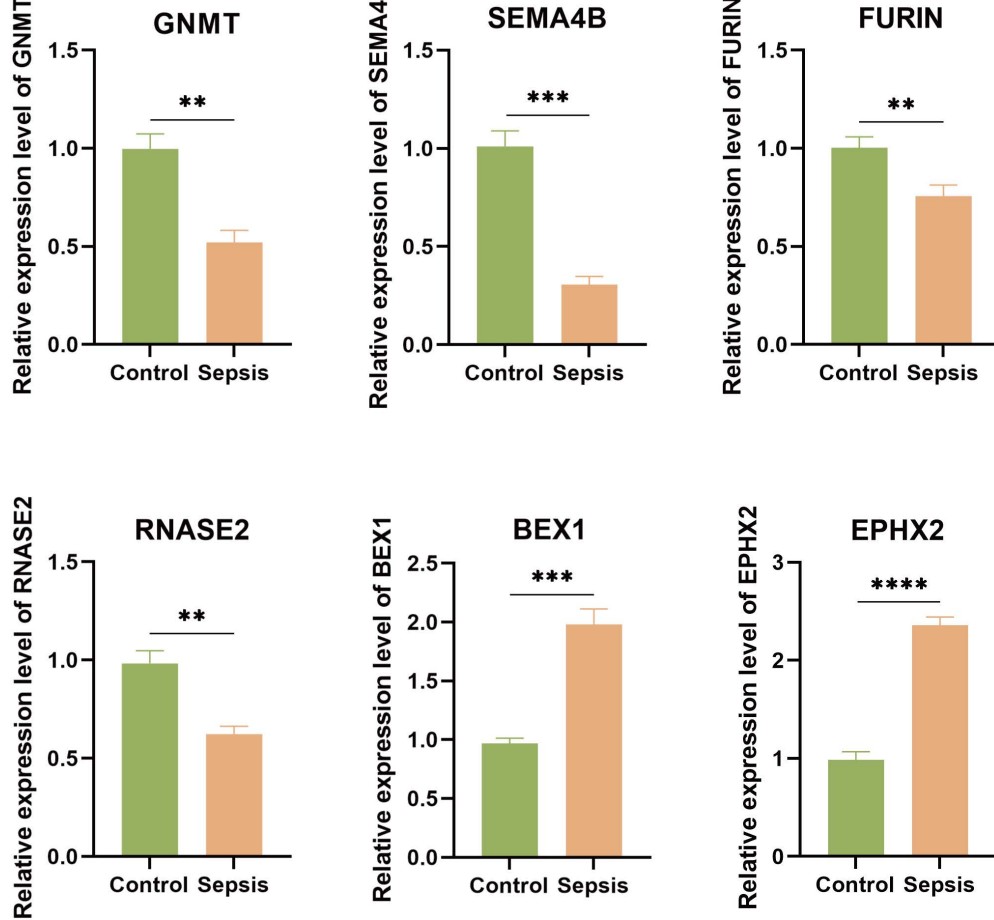

**Fig 8. The Relative Expression Levels of Six Key Genes in Sepsis Models.** The qRT-qPCR results demonstrated that the expression levels of GNMT, SEMA4B, FURIN, and RNASE2 in the sepsis group (n = 9) were significantly lower than those of the control group (n = 9), whereas the relative expression levels of BEX1, and EPHX2 were significantly higher. **p 0.01, ***p 0.001, ****p 0.0001, t-test based p-value.

## 4. Discussion

In clinical medicine, sepsis and HF are two severe pathological conditions with a complex and interrelated relationship. Sepsis is a multifactorial syndrome, typically triggered by infection, resulting in a systemic inflammatory response and multi-organ dysfunction [22]. HF, characterized by the heart's inability to pump sufficient blood to meet the body's meta-bolic demands, can arise from various etiologies, including coronary artery disease, hypertension, and cardiomyopathy [23]. Sepsis has a profound impact on patients with HF, particularly those with HF with reduced ejection fraction (HFrEF). Studies show that individuals with HFrEF admitted for sepsis exhibit higher in-hospital mortality and increased 90-day readmission rates compared to patients without HF [24]. Additionally, studies have shown that the incidence of HF is higher in sepsis patients and that HF can worsen the prognosis of sepsis. Patients with HF may experience a higher risk of death when experiencing sepsis [25]. These findings underscore the bidirectional relationship between sepsis and HF, emphasizing the need for effective predictive models and targeted interventions to improve outcomes in this vulnerable patient population.

HF in septic patients is primarily driven by the release of cytokines, mitochondrial dysfunction, and tissue hypoxia, resulting in myocardial cell injury and death. During sepsis, a series of structural and functional alterations occur in myocardial mitochondria, leading to disturbances in cardiac energy metabolism. Oxidative stress and nitrosative stress can cause changes in the lipid components of mitochondrial membranes [26]. Reactive oxygen species (ROS) directly attack the mitochondrial membrane lipids in cardiomyocytes, resulting in lipid peroxidative damage and compromising the integrity of the inner mitochondrial membrane. The myocardial depression induced by sepsis may also be a consequence of the direct toxicity of sepsis mediators on myocardial cells. Myocardial dysfunction resulting from sepsis is one of the main reasons for the high mortality associated with sepsis, as it can induce heart failure through the modulation of mitochondrial dysfunction, metabolic changes, cell death, and signaling pathways [27].

In the early stages of sepsis, inflammatory cells are recruited to the sites of injury, leading to endothelial dysfunction, vascular occlusion, and the release of pro-inflammatory factors, degrading enzymes, and ROS [28]. Mitochondrial dysfunction is primarily due to increased ROS levels that disrupt the homeostatic concentrations within mitochondria, resulting in various reversible and irreversible toxic modifications of biomolecules, such as protein carbonylation and lipid peroxidation [29]. Simultaneously, excessive ROS and nitric oxide (NO) can damage the structure of the mitochondrial respiratory chain and exacerbate ROS biosynthesis [30,31]. The excess ROS generated by the myocardium itself or by infiltrating inflammatory cells alters intracellular signaling cascades, leading to severe oxidative stress within the cells.

The metabolic demands of sepsis require considerable energy derived from fat metabolism, amino acid metabolism, glucose metabolism, and absorption of energy from cellular metabolism to regulate immune function [32,33]. AMPK is a classical cellular energy sensor that reconnects metabolism and maintains redox balance. The regulation of cellular mechanisms by AMPK is highly relevant to septic myocardial injury [34,35]. Notably, AMPK activators have been explored as a novel approach to alleviate inflammation related to ROS-associated diseases, including sepsis-induced injuries and cardiovascular diseases.

Furthermore, ferroptosis and pyroptosis contribute to the pathogenesis of sepsis. Iron molecules promote the aggregation of ferritin on cell membranes through the HO-1 pathway; however, the activation of ferritin by iron leads to increased iron export, causing iron overload [36,37]. By regulating the key molecule GSDMD, pyroptosis activates the NLRP3 inflammasome and caspase-1-dependent apoptosis, contributing to myocardial dysfunction in sepsis.

In this study, we developed and validated a predictive model for HF using sepsis-related gene signatures, demonstrating robust performance across independent datasets. Our model, based on six key genes, significantly outperformed individual biomarkers in both training and validation datasets. Furthermore, immune infiltration analysis revealed distinct patterns in HF patients, with more pronounced immune changes observed at higher risk score levels. These findings provide new insights into the molecular links between sepsis and HF, offering potential tools for early diagnosis and risk stratification. The model, based on six key genes including EPHX2, BEX1, RNASE2, FURIN, SEMA4B, and GNMT identified through elastic net regularization, consistently outperformed individual biomarkers, highlighting the advantage of integrating multiple gene signatures into a composite risk score. Among these key genes, EPHX2 serves multiple roles in HF, influencing disease progression and clinical outcomes by affecting pathways such as PI3K/AKT and GSK3β signaling pathways [38–40]. BEX1 acts as an RNA-dependent mediator, stabilizing and enhancing the expression of pro-inflammatory mRNAs, thereby contributing to the progression of cardiac disease [41]. In addition, Martens et al. demonstrated that BEX1 is both necessary and sufficient to counteract viral replication in dissociated primary cardiomyocytes and mouse embryonic fibroblasts. BEX1 can interfere with the expression of coxsackievirus B3 (CBV3) by regulating the expression of interferon-beta in infected cells. This antiviral effect may have broader applicability, extending beyond CVB3 to include other viruses, such as influenza A and Sendai virus [42]. RNASE2, associated with subclinical chronic inflammation, can predispose individuals to acute cardiac complications [43,44]. FURIN expression is often elevated in HF, promoting BNP maturation and intensifying cardiac stress responses [45,46]. Additionally, the product of the SEMA4B gene, Sema4D, has emerged as a critical player in the pathophysiology of HF [47]. The ROC curve analysis further confirmed

the model's reliability, showing superior performance in distinguishing HF patients from controls across both training and validation datasets. Together, these findings underscore the relevance of these key genes in cardiac dysfunction and highlight the potential of this multi-gene model for early diagnosis and personalized risk stratification in HF patients.

In our study, In high-risk HF patients, immune checkpoint genes—responsible for regulating immune exhaustion and preventing excessive immune activation—are expressed at lower levels. This suggests that in these patients, a lack of immune suppression may drive persistent immune activation and inflammation, exacerbating cardiac dysfunction and contributing to disease progression [48,49]. In contrast, the positive correlation between the risk score and HLA gene expression indicates enhanced antigen presentation capacity. Research suggests that the expression of HLA class I antigens may contribute to immune escape mechanisms within the heart, allowing cardiac cells to evade immune surveillance and thereby exacerbating the progression of HF [50].

Despite our predictive model has strong performance and has been validated on independent datasets, there are still some limitations. The observational design limits the interpretation of causal relationships, and reliance on publicly available datasets may introduce bias, emphasizing the necessity of prospective, multicenter validation. Furthermore, while the model provides valuable insights for early diagnosis and risk stratification, further improvement by integrating other omics data such as proteomics or metabolomics can enhance its predictive accuracy and clinical applicability. While the selected characteristic genes may serve as biomarkers for HF, they have not been fully validated in clinical samples such as serum, and there is insufficient experimental evidence. Although our research findings have to some extent revealed the relationship between sepsis and HF, and identified relevant biomarkers, our study relies on public databases and cannot represent the entire global community. In the future, a larger global database of samples will be needed for more in-depth exploration and validation.

## 5. Conclusion

To summarize, our study demonstrates the potential of integrated multi-omics analysis in identifying sepsis-related gene signatures that improve predictive modeling of heart failure. Our findings highlight the importance of these gene signatures in elucidating the pathophysiology of heart failure, providing a foundation for future research and clinical interventions aimed at early detection and management.

## Supporting information

**S1 Fig. Comparison of Immune Checkpoint Gene Expression Levels between High- and Low-Risk Groups.** *P 0.05, **p 0.01, ***p 0.001.
(TIF)

**S1 Table. Primers for qRT-PCR.**
(DOCX)

## Author contributions

**Conceptualization:** Fei Li.

**Data curation:** Yiping Lang, Tianyu Liang.

**Formal analysis:** Yiping Lang.

**Funding acquisition:** Fei Li.

**Investigation:** Yiping Lang, Tianyu Liang, Fei Li.

**Methodology:** Fei Li.

**Project administration:** Fei Li.

**Resources:** Fei Li.

**Software:** Yiping Lang.

**Supervision:** Fei Li.

**Validation:** Yiping Lang, Tianyu Liang.

**Visualization:** Yiping Lang, Tianyu Liang, Fei Li.

**Writing – original draft:** Yiping Lang, Tianyu Liang.

**Writing – review & editing:** Yiping Lang, Tianyu Liang, Fei Li.

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
