## [Decision Letter · Decision Letter 0]

PONE-D-24-53311Integrated Multi-Omics Analysis and Predictive Modeling of Heart Failure Using Sepsis-Related Gene SignaturePLOS ONE

Dear Dr. Li,

Thank you for submitting your manuscript to PLOS ONE. After careful consideration, we feel that it has merit but does not fully meet PLOS ONE’s publication criteria as it currently stands. Therefore, we invite you to submit a revised version of the manuscript that addresses the points raised during the review process.

We look forward to receiving your revised manuscript.

Kind regards,

Muhammad Iqhrammullah, Ph.D

Academic Editor

PLOS ONE

Journal Requirements:

3. We note that your Data Availability Statement is currently as follows: All relevant data are within the manuscript and in Supporting Information files.

Reviewers' comments:

Reviewer's Responses to Questions

**Comments to the Author**

1. Is the manuscript technically sound, and do the data support the conclusions?

Reviewer #1: Yes

Reviewer #2: Yes

Reviewer #3: Yes

2. Has the statistical analysis been performed appropriately and rigorously? 

Reviewer #1: Yes

Reviewer #2: Yes

Reviewer #3: Yes

3. Have the authors made all data underlying the findings in their manuscript fully available?

Reviewer #1: Yes

Reviewer #2: Yes

Reviewer #3: Yes

4. Is the manuscript presented in an intelligible fashion and written in standard English?

Reviewer #1: Yes

Reviewer #2: Yes

Reviewer #3: Yes

5. Review Comments to the Author

Reviewer #1: A review of the manuscript entitled “Integrated Multi-Omics Analysis and Predictive Modeling of Heart Failure Using Sepsis-Related Gene Signature”

1. The authors should improve their literature review by including recently published papers. Additionally, besides summarizing previous work in the topic, the authors should link their approach with previous studies by highlighting the state-of-the-art and gap in the literature.

2. (page 9): These sentences “HF is a prevalent cardiovascular disorder that impacts the health of millions worldwide…” also share the same idea with a study by Gusti et al. entitled “Association between admission blood sugar levels and length of stay among patients with acute heart failure: A cross-sectional study in Aceh, Indonesia” https://doi.org/10.52225/narrax.v1i2.75. Kindly include this reference to make the sentences stronger.

3. (page 22): This sentence “HF, characterized by the heart’s inability to pump sufficient blood to meet the body’s metabolic demands, can arise from various etiologies, including coronary artery disease, hypertension, and cardiomyopathy” will be more significant if it’s also supported by another relevant study. Please check https://pubmed.ncbi.nlm.nih.gov/39280297/

4. What is the conclusion of this study?

Reviewer #2: This is an interesting predictive model development study on Predictive Modeling of Heart Failure using Sepsis-related gene signature in China. The topic is interesting and the authors present a novel and well-defined methodology, and present the data adequately therefore I recommend the journal to publish this paper.

I have several major concerns and a few suggestions on how to improve the readability of the manuscript. General comments:

This study has demonstrated areas which are covered by this journal therefore publications is recommended. The main objective of the current review, according to the authors, is to provide reliable tools for early diagnosis and risk stratification. However, the authors have not robustly stated why is there a need of reliable tools for early diagnosis and risk stratification based on bioinformatics? Could this idea be implemented worldwide? This should also be addressed in the manuscript.

Specific comments:

1. Abstract:

- Abstract has been written well

2. Introduction:

- The author have argued well the reason why this study has to be conducted using valid and clear statement. However, robust explanation regarding the need for bioinformatics-based reliable tools for early diagnosis and risk stratification is still expected from the author, especially for the world population. The author could benchmark into several gene related studies.

- Please add more citations on key recent studies or relevant prior work in the field, such as studies on heart failure biomarkers or predictive modeling approaches?

- The author could improve elaborations on how this study could not only help identify HF risk in sepctic patients using reliable predictive models, but could impact in similar settings or other regions or countries, thus this explanation should be provided.

3. Methods

- The methods is too long, please simplify

- Please also elaborate robustly about what statistical analysis was done in every step in this study.

- “Additionally, we explored potential PPIs involving the model genes by identifying interacting proteins and calculating cumulative interaction scores based on multiple layers of supporting evidence.” � Please add citations

- If possible, please provide external validation methods

4. Results

- The authors have shown results clearly. However, is it possible to assess the risk factors with Odds ratio or any other effect size? Please add if possible.

- “ROC curve analysis demonstrated that the risk score achieved superior predictive performance for HF compared to any individual gene in the training dataset “ Please provide any numbers which supported this statement.

- If possible, please provide other parameter than just ROC curves. For instance AUC curves? (if possible)

- If possible, please add external validation results done in this study.

5. Discussion and Recommendations

- Discussion is too long, please shorten the discussion.

- While the clinical importance of sepsis and HF is well-explained, it may be useful to briefly mention why predictive modeling in this context is so crucial. Please elaborate more on the mechanisms underlying the bidirectional relationship, perhaps touching on immune dysregulation, metabolic pathways, or inflammatory processes.

- The interpretation of the findings is generally clear, but more explicit connections between the results and the broader clinical implications could strengthen the discussion. For instance, when introducing each key gene, consider highlighting how its specific role might influence treatment strategies or prognosis in sepsis-related HF.

- Please provide a bit more detail of specific metrics. For instance accuracy, sensitivity, or specificity. Please also provide a brief comparison to existing models (if available) to highlight how this model would improves compared current approaches.

- Are each gene’s role specifically in sepsis-related heart failure? Or Are these genes implicated in other cardiovascular diseases or other inflammatory conditions?

- If possible, please provide wide implementations and recommendations to improve readability of the paper

- Consider expanding on the clinical significance of immune checkpoint modulation. Could immune checkpoint inhibitors or immune modulation therapies potentially be explored as adjuncts in HF patients with sepsis?

6. Conclusion

- Conclusion is strong and well written. Please add what should be the clinical implications after this study.

7. Language

- Good

Reviewer #3: Overall, I think this manuscript merits publications. Great job! Although, there are some suggestions and questions for clarification.

Abstract

• The model development process is not clearly outlined—what were the inclusion criteria for key genes? How was performance assessed beyond ROC analysis?

• "superior performance compared to individual genes" lacks specificity. Provide quantitative results (e.g., AUC values) instead of qualitative statements.

• The claim that the model provides "valuable insights into molecular mechanisms" is broad and unsubstantiated without mentioning key pathways or biological processes.

• The abstract does not emphasize what is novel about this study compared to previous research. Is this the first study linking sepsis-related genes to HF prediction?

Introduction

• In depth explanation about HF itself is needed, please cite https://doi.org/10.1016/j.cjco.2023.08.006,
https://doi.org/10.1186/s43044-024-00558-3, and https://doi.org/10.1186/s43044-024-00580-5

• While sepsis-induced cardiac dysfunction is acknowledged, the introduction lacks a thorough discussion on why sepsis-related genes are relevant to predicting HF.

• The introduction should include a stronger rationale for focusing on sepsis-specific gene signatures rather than broader inflammatory or cardiac biomarkers.

• Emphasize the knowledge gap in existing HF predictive models and why sepsis-related genes might improve upon them.

• The shift from discussing clinical aspects of HF to bioinformatics lacks coherence. Clearly state why an integrative multi-omics approach is needed and how it surpasses existing predictive models.

Methods

• The manuscript does not explain why these specific sepsis datasets were chosen. Were alternative datasets available?

• It is unclear whether the HF datasets used for validation match the demographic/clinical characteristics of the sepsis datasets. A lack of harmonization could introduce biases.

• How was the effectiveness of ComBat assessed post-correction? Did the batch effect correction introduce data distortion or information loss?

• The PCA analysis is described, but it is unclear how much variance was explained by the principal components.

• What was the impact of dimensionality reduction on model performance? Were genes retained based solely on statistical separation, or was biological relevance considered?

• The manuscript does not state how PPI analysis contributes to the study’s predictive modeling. Was it used to refine gene selection, or was it merely exploratory?

• The choice of the ‘dynamicTreeCut’ algorithm is not justified. Were alternative clustering methods (e.g., WGCNA) considered?

• The manuscript states that similar KEGG terms were merged, but this lacks transparency. What criteria were used for merging?

• The results should specify which pathways were enriched and discuss their relevance to HF. A simple list of pathways is not informative without interpretation.

• How were genes selected for inclusion in the final model? Was feature selection performed?

• The manuscript states that batch effects were corrected before model training, but does not clarify whether this was done separately for training and test sets.

• The manuscript mentions leave-one-study-out validation but does not clarify whether performance remained stable across different training subsets.

• Were genes selected based on statistical significance, biological relevance, or predictive importance in the model?

• ROC analysis alone is insufficient to assess model utility. Metrics such as precision-recall curves, calibration plots, or decision curve analysis should be provided.

Results

• The section describes overlaps in gene sets but does not elaborate on their biological significance in sepsis or heart failure. Are these overlapping genes known to play a role in inflammation or cardiac dysfunction?

• While PCA and z-score normalization are mentioned, it is unclear how batch-corrected expression values were standardized across datasets. Was cross-validation performed to ensure consistency?

• The heatmap shows inter-patient variability, but no quantitative measures (e.g., variance analysis) are provided to assess whether specific patient subgroups contribute disproportionately to the observed heterogeneity.

• The GO and KEGG results mention enriched pathways but fail to provide enrichment scores, p-values, or FDR-adjusted values. The significance of these pathways remains unclear without statistical support.

• While the model was tested on independent datasets, there is no mention of performance metrics beyond "effective distinction." Metrics such as AUC-ROC, sensitivity, specificity, or F1-score should be reported for evaluation.

• It is unclear whether cross-validation was performed to ensure the generalizability of the model. Were hyperparameters optimized using an independent validation set, or was the model trained and tested on overlapping samples?

• How does it compare to existing HF biomarkers?

• The text states that immune checkpoint genes are negatively correlated with risk score, but does not specify which immune checkpoints are affected. A more detailed breakdown is required.

Discussion

• The discussion largely reiterates known relationships between sepsis and HF but does not sufficiently integrate the study’s unique contributions. How do the identified genes improve current understanding of sepsis-induced HF?

• The discussion claims that immune infiltration changes are associated with HF risk but does not explore the underlying pathways responsible for these alterations. Which immune cell types are most affected? What cytokines or signaling pathways are implicated?

• While the predictive model is statistically validated, there is no discussion on how it could be used in clinical practice. Are there potential biomarkers that could be tested in blood samples? How does this model compare to traditional HF risk scoring systems?

• Were there any outliers or misclassified samples that affected the model’s performance?

6. PLOS authors have the option to publish the peer review history of their article (what does this mean? ). If published, this will include your full peer review and any attached files.

**Do you want your identity to be public for this peer review?** For information about this choice, including consent withdrawal, please see our Privacy Policy .

Reviewer #1: No

Reviewer #2: **Yes: ** Ayers Gilberth Ivano Kalaij

Reviewer #3: No

---

## [Author Response · Author response to Decision Letter 1]

17 Apr 2025

Dear Editor and reviewers:

Thank you very much for your detailed constructive and active comments on our manuscript.

According to each comment from reviewers, we have revised the manuscript carefully and made corresponding changes point by point. All the changes including the text and tables have been added and labeled in yellow in the revised manuscript according to this response. The list of changes has been itemized and explained in the "Response to Reviewers" file.

Thanks again for your constructive suggestions. Thanks for your kind consideration of our article again. If you have any questions about the manuscript, please feel free to contact me.

Sincerely yours,

Fei Li

Email: lifeilifeiicu@163.com

---

## [Decision Letter · Decision Letter 1]

Integrated Multi-Omics Analysis and Predictive Modeling of Heart Failure Using Sepsis-Related Gene Signature

PONE-D-24-53311R1

Dear Dr. Li,

We’re pleased to inform you that your manuscript has been judged scientifically suitable for publication and will be formally accepted for publication once it meets all outstanding technical requirements.

Kind regards,

Muhammad Iqhrammullah, Ph.D

Academic Editor

PLOS ONE

Additional Editor Comments (optional):

Reviewers' comments:

Reviewer's Responses to Questions

**Comments to the Author**

1. If the authors have adequately addressed your comments raised in a previous round of review and you feel that this manuscript is now acceptable for publication, you may indicate that here to bypass the “Comments to the Author” section, enter your conflict of interest statement in the “Confidential to Editor” section, and submit your "Accept" recommendation.

Reviewer #3: All comments have been addressed

2. Is the manuscript technically sound, and do the data support the conclusions?

Reviewer #3: Yes

3. Has the statistical analysis been performed appropriately and rigorously? 

Reviewer #3: Yes

4. Have the authors made all data underlying the findings in their manuscript fully available?

Reviewer #3: Yes

5. Is the manuscript presented in an intelligible fashion and written in standard English?

Reviewer #3: Yes

6. Review Comments to the Author

Reviewer #3: Accept. I think the authors have done the revision in a concise and direct manner. I wish them goodluck.

7. PLOS authors have the option to publish the peer review history of their article (what does this mean? ). If published, this will include your full peer review and any attached files.

**Do you want your identity to be public for this peer review?** For information about this choice, including consent withdrawal, please see our Privacy Policy .

Reviewer #3: No
